# Discrete choice experiment versus swing-weighting: A head-to-head comparison of diabetic patient preferences for glucose-monitoring devices

**Chiara Whichello**[1,2,3]*, **Ian Smith**[2,4], **Jorien Veldwijk**[2,3], **G. Ardine de Wit**[4], **Maureen P. M. H. Rutten- van Molken**[2,3], **Esther W. de Bekker-Grob**[2,3]

**1** Evidera, London, United Kingdom, **2** Erasmus Choice Modelling Centre, Erasmus University Rotterdam, Rotterdam, The Netherlands, **3** Erasmus School of Health Policy & Management, Erasmus University Rotterdam, Rotterdam, The Netherlands, **4** Julius Centre for Health Sciences and Primary Care, University Medical Centre Utrecht, Utrecht University, Utrecht, The Netherlands

* chiara.whichello@evidera.com

**Data Availability Statement:** The participants in this study have not consented to deposition of the data. As these data contain sensitive information,

## Abstract

### Introduction

Limited evidence exists for how patient preference elicitation methods compare directly. This study compares a discrete choice experiment (DCE) and swing-weighting (SW) by eliciting preferences for glucose-monitoring devices in a population of diabetes patients.

### Methods

A sample of Dutch adults with type 1 or 2 diabetes (n = 459) completed an online survey assessing their preferences for glucose-monitoring devices, consisting of both a DCE and a SW exercise. Half the sample completed the DCE first; the other half completed the SW first. For the DCE, the relative importance of the attributes of the devices was determined using a mixed-logit model. For the SW, the relative importance of the attributes was based on ranks and points allocated to the 'swing' from the worst to the best level of the attribute. The preference outcomes and self-reported response burden were directly compared between the two methods.

### Results

Participants reported they perceived the DCE to be easier to understand and answer compared to the SW. Both methods revealed that cost and precision of the device were the most important attributes. However, the DCE had a 14.9-fold difference between the most and least important attribute, while the SW had a 1.4-fold difference. The weights derived from the SW were almost evenly distributed between all attributes.

### Conclusions

The DCE was better received by participants, and generated larger weight differences between each attribute level, making it the more informative method in our case study. This

all relevant data are therefore only available upon request. A de-identified minimal data set is available to researchers who meet the criteria for access to confidential data via request to the Senior Registrar Clerk at Erasmus University Rotterdam, Esther de Bekker-Grob (debekker-grob@eshpm. eur.nl). Interested researchers may also contact the corresponding author to request the data used for the analyses in this paper.

**Funding:** The Patient Preferences in Benefit-Risk Assessments during the Drug Life Cycle (PREFER) project has received funding from the Innovative Medicines Initiative 2 Joint Undertaking under grant agreement No 115966. This Joint Undertaking receives support from the European Union's Horizon 2020 research and innovation programme and EFPIA. This text and its contents reflect the PREFER project's view and not the view of IMI, the European Union or EFPIA.

**Competing interests:** The authors have declared that no competing interests exist.

method comparison provides further evidence of the degree of method suitability and trustworthiness.

## Introduction

The integration of patient preferences into decision-making is becoming progressively more important throughout the medical product lifecycle (MPLC) [1]. Projects such as IMI-PRE-FER [2] and the MDIC (Medical Device Innovations Consortium) [3] are promoting the importance of patient preference information in benefit-risk assessments, while the National Institute for Health and Care Excellence (NICE) is establishing patient preference research partnerships [4]. There is consensus among industry, regulatory, and health technology assessment (HTA) stakeholders that patient preference information would be beneficial when informing benefit-risk assessments throughout the MPLC [2, 5]. This includes the selection of endpoints in early clinical development, to inform regulatory benefit-risk assessments, and to be submitted alongside reimbursement dossiers for HTA appraisal [6].

Different preference measurement techniques exist for specific decision-making contexts. These contexts reflect situations where this patient preference information have high value, such as when there are multiple, alternative treatments with very different benefit-risk profiles [7]. It is vital that decision-makers and researchers select the most appropriate methods suitable for preference-sensitive contexts. However, there is a lack of guidance in current literature regarding the suitability of different patient preference elicitation methods for different situations [7, 8]. A recent empirical comparison has identified discrete choice experiments (DCE) and swing-weighting (SW) as being among the most promising methods likely to meet decision-makers' needs throughout the MPLC [8].

DCEs are derived from random utility theory (RUT), and assume that a healthcare intervention can be represented by its characteristics (also called attributes) [9, 10]. The relative importance of these attributes can be determined by presenting a series of hypothetical choice tasks, and asking for participants' preferred option. The relative weights for each attribute and attribute-level can be derived statistically [11]. DCE outcomes can be used to answer a number of different research questions including trade-off quantification, the willingness-to-pay for different alternatives, and expected uptake rates [12].

SW determines the relative importance based on the improvement of an attribute from its worst state to its best state [13, 14]. Each attribute is first ranked by participant reflecting the importance of this 'swing' from the worst to best level, then points are assigned to each ranking during what is referred to as 'point allocation'[15, 16]. SW also assumes that a participant's utility can be summarized by an explainable value where an individual is always assumed to select an alternative with a higher utility.

Both of these methods can be used to assess the relative value that different attributes and attribute-levels have for the participant. However, whether different methods lead to the same conclusions, when answering the same research question, is a research topic in need of investigation [16]. Literature comparing DCE and SW is lacking [14, 17], although both methods are increasingly used in healthcare to empirically evaluate the relative desirability of treatment options or attributes [3, 13]. Rating methods, such as swing-weighting, are often regarded as a simpler approach to eliciting patient preferences since they do not force simultaneous trade-offs between multiple attributes [15]. However, other health economists state that direct pairwise comparisons in a DCE are easier for patients than a direct numerical assessment of

relative value present in SW [14]. Therefore, the aim of this study is to compare the performance and results of DCE and SW in a common preference context through empirical research.

## Methods

This study compares the DCE and SW in the context of preferences for glucose-monitoring technologies in diabetes patients. Recent advancements in glucose monitoring technology have led to the introduction or new devices to the consumer market such as continuous glucose monitors (CGMs) and flash glucose monitors (FGMs) [18]. These devices are less invasive and more user-friendly than the more commonly used fingerprick-test, which involves direct testing of the blood by lancing the finger multiple times per day to extract a blood sample. The functions, features, and associated costs of CGMs and FGMs vary greatly between devices resulting in a preference sensitive situation [19]. Therefore, the benefit-risk trade-offs affecting a patient's decision for selecting a glucose-monitoring device deserves closer investigation.

### Attributes and level development

The development of attributes and attribute-levels used to describe the glucose monitoring devices for both the DCE and the SW was conducted in three steps (see Fig 1). In step 1, a scoping literature review was conducted in PubMed to identify relevant attributes of glucose-monitoring devices and develop an interview guide. In step 2, semi-structured interviews were conducted with Type 1 and 2 diabetes patients (n = 19), clinicians (n = 5), patient organization representatives (n = 2), and pharmaceutical industry representatives involved in glucose monitoring device development (n = 4) which resulted in an initial list of 12 relevant attributes. In step 3, the list of 12 attributes were rated and reduced according to relevance, completeness, non-redundancy, operationality, and preferential independency by the research team. Five attributes were removed based on failing these criteria. Subsequently, seven attributes were

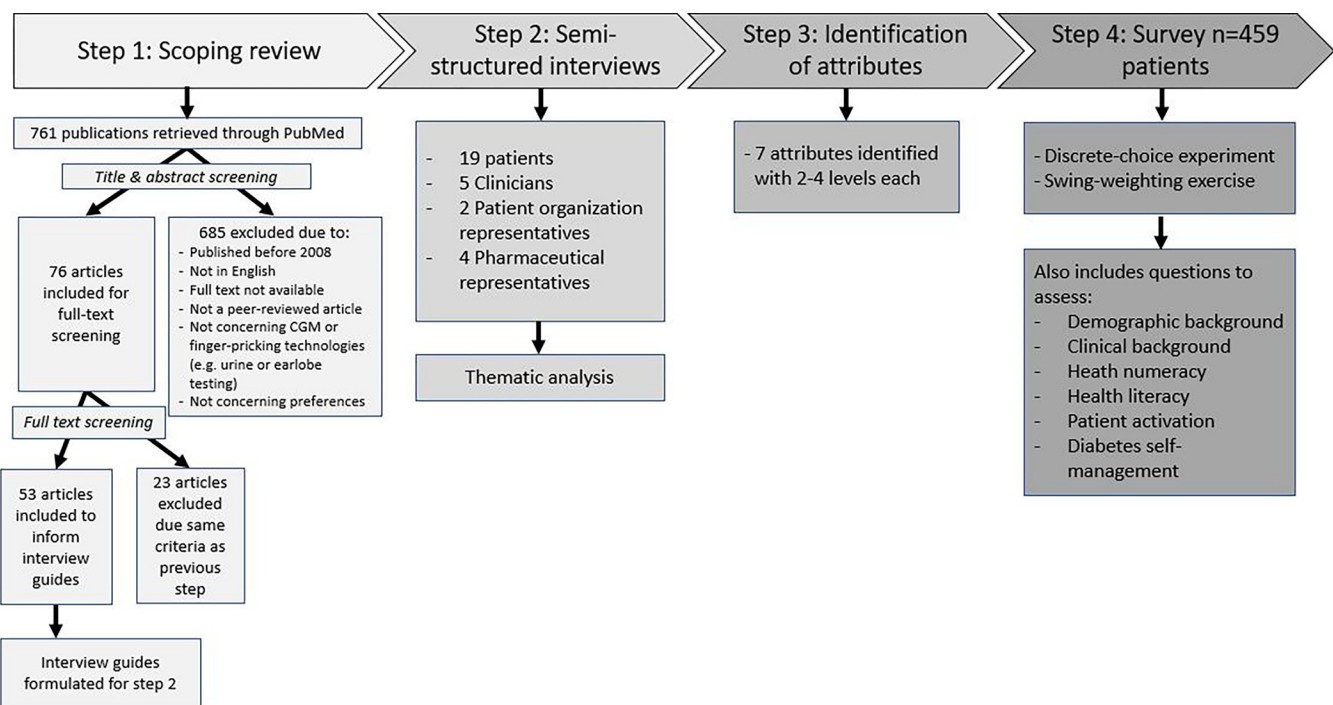

**Fig 1. Methodology steps for developing the survey.**

**Table 1. Attributes and levels for the discrete choice experiment and swing-weighting.**

| Attributes | Level 1 | Level 2 | Level 3 | Level 4 |
|---|---|---|---|---|
| **Precision compared to fingerpricking** [a] | Less accurate than fingerpricking (higher or lower by 0.6)* | Less accurate than fingerpricking (higher or lower by 0.3) | Accurate as fingerpricking* | — |
| **Average number of fingerpricks per day** [b] | 4* | 2 | 0* | — |
| **Effort to check** [c] | High effort: you need to measure your glucose levels yourself* | Moderate effort: you scan a sensor to check glucose levels | Low effort: glucose levels automatically sent to you* | — |
| **Probability of getting skin irritation or redness** [d] | 35% chance of skin irritation or redness* | 20% chance of skin irritation or redness | 5% chance of skin irritation or redness | No chance of skin irritation or redness* |
| **Monthly costs** [e] | €250* | €175 | €100 | €25* |
| **Glucose information** [f] | Current Glucose level* | Current Glucose level and arrow | Current Glucose level and a graphic of your level trends over the day* | — |
| **Alarms** [g] | No* | Yes* | — | — |

* Level included in SW (method only contains highest and lowest levels within attributes)

(a-g) Attribute explanations as presented to patients

Some glucose monitors are more precise than others. Fingerpricking is generally regarded as the most accurate way to measure glucose levels. Measurements from devices that use sensors can be just as accurate, but can also be less accurate than fingerpricking, especially if your glucose levels are very high or very low. For example, if your glucose level is 6 mmol/L and you measure it with a device that is off by 0.6 mmol/L, then this device can say your glucose is anywhere from 5.4 to 6.6 mmol/L

This is how many times you would need to do a fingerprick-test each day on an average day. This number could be higher on days when you feel the need to test more often like when you're sick, but we want you to picture an average day. Sometimes, this is your only method of measuring your glucose levels. Or, you might need to do fingerprick-tests to confirm the levels from another device

This means how much effort you need to give to check your blood glucose levels. High effort checking means you need to stop what you're doing and concentrate on measuring your levels. You need to wash your hands, get out your device equipment, prick your finger, put blood on a strip, check the results, and then clean everything up. Moderate effort checking means you need to get out a small device and use it to scan the sensor on your body to obtain your glucose levels. Low effort checking means your glucose levels are automatically sent to a device which you can view at any time. This could be a dedicated glucose device, your phone, or a smartwatch. You don't need to do anything to have your blood glucose levels sent through, just look at the device to check.

A chance of skin irritation or redness around a sensor means a redness or itchy rash on the skin around or under the sensor. This is similar to having an itchy allergic reaction and can be rather uncomfortable or irritating. The sensor will need to be removed and replaced in a different spot. This skin irritation and redness usually lasts until after the sensor is replaced. Not all sensor have this side effect so chances of getting the side effect can differ per device. If a device gives you a 15% chance, this means that 15 out of a 100 people who get this device experience skin irritation and redness while 85 out of a 100 people do not experience this.

This means how much money you need to pay out-of-pocket per month in order to check your blood glucose. Please note that this is money that is not reimbursed by your insurance. This could be money needed to pay for devices, sensors, or strips used.

This means how your glucose levels are presented to you. This information could be only your current glucose level (you only see a digital number like 8.3 mmol/L). This could be your current glucose level with an arrow showing how your blood glucose is changing as compared to your previous measurement (increasing, decreasing, stable). Or, it could show your current glucose level with a graphic of your blood glucose levels over the day.

Your device will give you a beeping alarm (like a phone notification) any time your blood glucose levels are (getting) too high or too low.

selected for the DCE and SW (Table 1). The levels used to describe the attributes were based on real-world data [20], representing the most common types of glucose monitoring devices, including CGMs and FGMs [21–23]. The DCE incorporates all levels, while the methodology of the SW examines only the 'swing' from the lowest level to the highest level. The attributes and levels were presented identically in both methods in order to make accurate comparisons and avoid framing effects. The draft questionnaire was pre-tested during six 'think-aloud' tests, checking for comprehensibility and clarity.

## Design

**Discrete choice experimental design.** NGene 1.0 [24] software was used to develop a Bayesian D-efficient design, consisting of three blocks of 12 choice tasks. Each contained three

alternatives (i.e. profiles) with seven attributes of varying levels; two alternatives represented hypothetical glucose-monitoring devices and one represented the fingerpricking test. Participants were given two 'warm-up' DCE choice-tasks before the main exercise in order to ensure comprehension. The questionnaire was tested in a pilot of 99 participants in order to retrieve priors, which informed the design of the final DCE to optimise statistical efficiency.

After the pilot test of the DCE, the DCE design with three alternatives per choice task was substituted by a "best-best" or a so-called 'dual response' DCE [25]. In this task participants were asked which of the two hypothetical device alternatives they would prefer, either 'Device A' or 'Device B'. Then a follow-up task asked if they would prefer the hypothetical device chosen or a standard fingerprick-test (see Appendix I in S1 File for an example choice task). This design improves data quality by reducing the chance that participants default to the standard opt-out in order to decrease the burden of evaluating the alternatives, while maintaining a realistic decision context in which opting for the fingerprick-test is a reasonable option [26].

**Swing-weighting design.** The SW contained two parts. First, participants were asked to rank the seven attributes based on how they would prioritise improving each swing of an attribute-level from its worst to its best state (see Appendix II in S1 File for an example exercise). The seven attributes were listed randomly for each participant in order to prevent an intrinsic top-down ranking bias. Thereafter, participants were asked to allocate points, from 0 to 100, to each of the swings relative to their first choice which was automatically allocated 100 points [16, 28]. For instruction, participants were informed that if they allocated an attribute 50 points, this indicated they thought improving its state was half as important as their first ranked attribute-level. If participants attempted to allocate more points to a lower-ranked attribute that a higher-ranked attribute, they were presented with a pop-up message drawing attention to this action and ask them to confirm that they wish to proceed with the allocation.

**Questionnaire.** The questionnaire was online and self-administered. After providing online informed consent, participants received information on the meaning of all the attributes and levels, and then completed demographic questions. All respondents completed both the DCE and SW exercises, but the order was randomised with half of respondents seeing the DCE first and the other half seeing the SW first. Each exercise was followed by debriefing questions related to the ease of understanding the exercise and ease of completing the exercise. Respondents answered on a Likert scale from 1 to 6; 1 being the most difficult and 6 being the easiest. In between the DCE and SW, patients answered questions about their medication and glucose monitoring devices they currently used to control their diabetes, and the frequency of use. At the end of the questionnaire, health literacy and numeracy were assessed using the validated questions of the Shortened Subjective Numeracy Scale (SNS-3) [27] and the Brief Health Literacy Screener (Chew Items) [28].

Members of an online panel who are adult Dutch residents with type 1 or type 2 diabetes were invited to complete the survey. Diabetes diagnosis was self-reported, with no restrictions on type 1 or 2. This study (Reference number WAG/mb/19/045208) was granted approval by the Medical Research Ethics Committee, UMC Utrecht. More information about ethical approval can be found in Appendix III in S1 File.

## Statistical analysis

**Discrete choice experiment analysis.** The DCE was analysed by combining the outcomes of both best-best tasks into one task comparing all three alternatives (Device A versus Device B versus the fingerprick-test). The outcome of the second best-best task (hypothetical device chosen or fingerprick-test) was used to determine the participant's choice for use in the final model. Observations were analysed in NLOGIT [29] by a latent-class model and a mixed-logit

model [30]. Based on model fit, the mixed-logit was the model best suited to the data and the following utility function was used for the final analyses:

$$
\begin{aligned}
V_{Device\ A} = \beta_0 &+ \beta_1 * precision_{0.3} + \beta_2 * precision_{0.6} + \beta_3 * pricks\ per\ day_{2x} + \beta_4 * effort_{moderate} \\
&+ \beta_5 * skin\ irritation_{20\%} + \beta_6 * skin\ irritation_{35\%} + \beta_7 * monthly\ costs_{€100} \\
&+ \beta_8 * monthly\ costs_{€175} + \beta_9 * monthly\ costs_{€250} + \beta_{10} * information_{arrow} \\
&+ \beta_{11} * information_{trendline} + \beta_{12} * alarms_{none} \qquad\qquad\qquad \text{Eq 1}
\end{aligned}
$$

$$
\begin{aligned}
V_{Device\ B} = &\ \beta_1 * precision_{0.3} + \beta_2 * precision_{0.6} + \beta_3 * pricks\ per\ day_{2x} + \beta_4 * effort_{moderate} \\
&+ \beta_5 * skin\ irritation_{20\%} + \beta_6 * skin\ irritation_{35\%} + \beta_7 * monthly\ costs_{€100} \\
&+ \beta_8 * monthly\ costs_{€175} + \beta_9 * monthly\ costs_{€250} + \beta_{10} * information_{arrow} \\
&+ \beta_{11} * information_{trendline} + \beta_{12} * alarms_{none} \qquad\qquad\qquad \text{Eq 2}
\end{aligned}
$$

$$
V_{Fingerprick} = \beta_{13} \qquad\qquad\qquad \text{Eq 3}
$$

where $V$ represents the total relative utility for an alternative where $\beta_1$ to $\beta_{12}$ are coefficients reflecting the relative importance of each attribute or attribute-level. $\beta_{13}$ is an alternative specific constant reflecting the respondents' preference for the fixed alternative of the fingerprick-test over Device B. $\beta_0$ is a constant term which identifies the respondent's preferences for Device A over Device B, reflecting a left-right bias (i.e. favouring the left option in case the coefficient is significant and has a positive sign). All attributes and attribute levels were included as random parameters, with a normal distribution, accounting for any heterogeneity in the preferences for those attributes. Robust outcomes were generated by applying 14,000 Halton draws.

The mean of the individual uptake probabilities ($\bar{P}$) was determined by estimating the individuals'($i$) utility of a device ($V_i$) compared to the individuals' utility of the fingerprick alternative ($W_i$), calculating the probability of this choice, and averaging this across all individuals:

$$
\bar{P} = \frac{1}{n} \sum_{i=1}^{n} \frac{e^{V_i}}{e^{V_i} + e^{W_i}} \qquad\qquad\qquad \text{Eq 4}
$$

where $i = 1$ represents the index of summation and $n$ is the total sample size. Effects coding was used, meaning the reference category is coded as -1, which sums the attribute-level coefficients in each category to zero.

**Swing-weighting analysis.**   The SW analysis was conducted by examining each participant's point allocation for each attribute-level improvement relative to the total number of points allocated. Then, the weighted average of each attribute was calculated across the entire participant sample via Eq 5:

$$
\bar{S}(a) = \frac{1}{n} \sum_{i=1}^{n} \frac{x_{i,a}}{\sum_{j=1}^{7} x_{i,j}} \qquad\qquad\qquad \text{Eq 5}
$$

where $\bar{S}(a)$ represents the average relative preference score of an attribute ($a$), $x_{i,a}$ is the points allocated to the attribute by individual $i$, $n$ is the total number of participants, $j$ is the index of summation for each attribute, and $i$ is the index of summation for each individual.

**Comparison of DCE and SW.**   The two methods were compared in two ways. First, self-reported feedback from participants indicating how easy the method was to understand and answer was used to compare the methods. These results were stratified by the method that was completed first, health literacy, and numeracy. Drop-out rates during the completion of the exercises were also compared as a proxy for participant burden.

Second, a comparison of how important each attribute was reported to be using each method was examined by looking at the proportion of preference for one attribute compared to the summed preferences for all attributes. For the DCE, this involved examining the absolute difference between the best level coefficients and the worst level divided by the sum of all these differences across the attributes. For the SW, one attribute's weight was calculated as a percentage of the total summed attributes' weights (as shown in Eq 5). The relative weights for both methods, reflected as a proportional percentage, were then directly compared.

**Sensitivity analyses.** Two sensitivity analyses were conducted. The first sensitivity analysis was conducted by comparing the weights derived from the point allocation of the SW against weights derived from the ranking portion of the SW calculated using the rank order centroid (ROC) method [15]. The ROC assigns relative weights for each attribute based on the order they were ranked, as defined by Roberts and Goodwin [31]. The proportional ROC weights were also compared against the proportional DCE weights.

Secondly, in order to determine whether there were significant differences in attribute rankings between the methods, the respondent-level ranking of the attributes in the DCE and the SW were identified by determining how each individual participant ranked the attributes, either by participant rankings for the SW, or a ranked marginal utility for the DCE method. These individual ranking were then compared using a (generalised) ordered logit model. For the SW, these were individual-specific ranking outputs derived from Step 1 of the SW method, ranking the attributes from 1 to 7. For the DCE, the individual-specific rankings of each attribute were determined by first examining the patient uptake rates for the most preferred device (high precision, zero fingerpricks, low effort, low skin irritability, 25 euro, plain information, no alarm) replicating the same device hypothetically created during the SW exercise (i.e. the swing from 'worst' to 'best' attributes). As detailed above, the systematic utility is defined as an additive function consisting of marginal utilities. Therefore, the coefficients for each corresponding attribute-level for each individual were then ranked from lowest to highest and given a corresponding value (1 to 7). Each participant's DCE rankings were then compared against their SW rankings through the ordered logit model, and determining the probability that an attribute had of being ranked 1–7 in either the DCE or the SW.

## Results

### Participants' characteristics

A total of 500 participants completed the survey. Participants who completed the survey faster than 70% of the mean response time were excluded, leaving a sample of 459 respondents (Appendix IV in S1 File). Furthermore, 233 participants completed the DCE first, while 226 participants completed the SW first.

The mean age of all respondents was 51 years old, with a near even split between male and female respondents. Twenty-seven percent of the total sample reported having diabetes Type 1, 69.1% reported having Type 2. Approximately 18.3% of all respondents already used a CGM or FGM, and 54.4% used fingerpricking. About 93.3% reported a "high" or "intermediate" level of education (Appendix IV in S1 File).

### Comparing DCE and SW results

**Feedback comparison.** Overall, the DCE was reported by participants to be both easier to understand and to complete, compared to the SW (Fig 2). This was true regardless of which method was completed first. Averaged scores for both ease of understanding and ease of answering the DCE were significantly higher (mean = 4.71, s.d. = 1.38; mean = 4.60, s.d. = 1.36, respectively) than ease of understanding and ease of answering the SW (mean = 3.85,

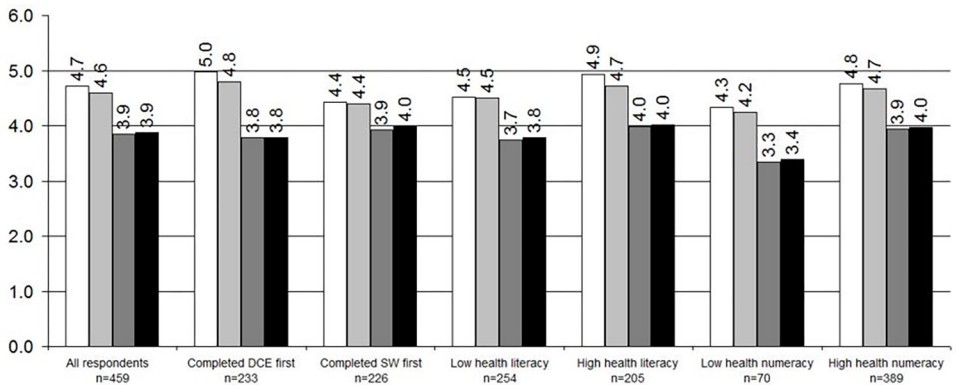

**Fig 2. Feedback scores from respondents completing the discrete choice experiment (DCE) and swing-weighting (SW).** *White* = Ease of understanding the *DCE; Light grey* = Ease of answering the DCE; *Dark grey* = Ease of understanding the SW*; Black* = Ease of answering the SW; Respondents answered on a scale from 1 to 6, 1 being the most difficult and 6 being the easiest; Low health numeracy scored below 9.83 (the mean) on the SNS-3; High health numeracy scored above 9.83 (the mean) on the SNS-3; Health literacy questions are scored 1–5 with the middle question inversed—Low health literacy identified by a score of >3 on any item; High health literacy scored <2 on any item (see Louis et al, 2016 [32]).

s.d. = 1.68; mean = 3.88, s.d. = 1.61, respectively). Both high-literacy and low-literacy participants rated the DCE higher than the SW, as did high-numeracy and low-numeracy participants.

Drop-outs were higher during the SW (n = 165) than the DCE (n = 101), regardless of the order of exercises. Of these, 143 first exercise drop-outs (did not proceed to their second exercise) occurred during the SW, compared to 93 during the DCE.

**Discrete choice experiment results.** The results of the mixed-logit (Table 2) showed significant estimates for all attribute-levels except for medium precision (0.3 mmol/L), glucose information (information-only), and glucose information with an arrow. Negative coefficients for the attribute-levels indicate that these would not be preferred features in a glucose-monitoring device, relative *to the mean* attribute effect. Higher monthly costs were associated with a lower willingness to choose the device. High precision (as accurate as fingerpricking) was strongly preferred over lower precision levels. Respondents generally preferred the fingerprick alternative over either device alternatives presented. However, the model showed significant heterogeneity in respondents' preferences for the constant as well as the other attributes. There also was a slight left-right bias detected.

The predicted uptake rates ranged from 65.9% for the most preferred device (high precision, zero fingerpricks, low effort, low skin irritability, 25 euro, plain information, no alarm) to 10.5% for the least preferred device (low precision, two fingerpricks, moderate effort, high skin irritability, 250 euro, an arrow, an alarm). Individual uptake probabilities did not vary significantly between individuals who saw the DCE first (67.4% for most-preferred device; 10.1% for least-preferred device) and those who saw the SW first (64.3% for most-preferred device; 11.0% least-preferred device).

**Swing-weighting results.** In general, respondents found cost to be the most important attribute with a mean relative weight of 0.17 (s.d. = 0.13), followed by precision (mean = 0.16; s.d. = 0.12) (see Table 3). The least important attribute was an alarm (mean = 0.12; s.d. = 0.11). These weights did not vary significantly between individuals who saw the DCE first or the SW first (the difference in mean was <0.02 for all attributes). There was little difference in relative weights given to the seven attributes, with all of the weights being almost evenly distributed across the attributes.

**Table 2. Attribute-level estimates for the discrete choice experiment mixed-logit model.**

| Attribute | Levels | | Estimate | p-value | S.E. |
|---|---|---|---|---|---|
| Precision compared to fingerpricking | Accurate as fingerpricking (ref) | Mean | 0.484 | * | 0.291 |
| | | S.D. | 0.762 | *** | 0.006 |
| | 0.3 | Mean | 0.043 | | 0.046 |
| | | S.D. | 0.087 | | 0.170 |
| | 0.6 | Mean | -0.527 | *** | 0.068 |
| | | S.D. | 0.757 | *** | 0.079 |
| Average number of fingerpricks per day | 0 times per day (ref) | Mean | 0.313 | *** | 0.068 |
| | | S.D. | 0.532 | *** | 0.003 |
| | 2 times per day | Mean | -0.313 | *** | 0.045 |
| | | S.D. | 0.532 | *** | 0.056 |
| Effort to check | Low (ref) | Mean | 0.165 | *** | 0.045 |
| | | S.D. | 0.231 | *** | 0.003 |
| | Moderate | Mean | -0.165 | *** | 0.033 |
| | | S.D. | 0.231 | *** | 0.058 |
| Probability of getting skin irritation or redness | 5% (ref) | Mean | 0.425 | *** | 0.059 |
| | | S.D. | 0.373 | *** | 0.008 |
| | 20% | Mean | -0.091 | * | 0.050 |
| | | S.D. | 0.011 | | 0.139 |
| | 35% | Mean | -0.334 | *** | 0.056 |
| | | S.D. | 0.373 | *** | 0.088 |
| Monthly costs | €25 (ref) | Mean | 1.728 | *** | 0.096 |
| | | S.D. | 1.878 | *** | 0.019 |
| | €100 | Mean | 0.325 | *** | 0.063 |
| | | S.D. | 0.243 | | 0.162 |
| | €175 | Mean | -0.128 | * | 0.067 |
| | | S.D. | 0.447 | *** | 0.113 |
| | €250 | Mean | -1.925 | *** | 0.139 |
| | | S.D. | 1.808 | *** | 0.125 |
| Glucose information | Information only (ref) | Mean | -0.133 | | 0.147 |
| | | S.D. | 0.108 | *** | 0.018 |
| | Arrow | Mean | 0.022 | | 0.049 |
| | | S.D. | 0.055 | | 0.142 |
| | Trendline | Mean | 0.111 | ** | 0.049 |
| | | S.D. | 0.094 | | 0.157 |
| Alarms | Yes (ref) | Mean | 0.151 | *** | 0.247 |
| | | S.D. | 0.348 | *** | 0.003 |
| | No | Mean | -0.151 | *** | 0.036 |
| | | S.D. | 0.348 | *** | 0.051 |
| Alternative specific constant for fingerprick-test† | | Mean | 0.949 | *** | 0.287 |
| | | S.D. | 5.089 | *** | 0.321 |
| Alternative specific constant indicating left-right bias | | Mean | 0.359 | *** | .070 |
| | | S.D. | 0.682 | ** | .103 |

* indicates p < 0.1

** indicates p < 0.05

** indicates p <0.01; S.D. indicates standard deviation; ref indicates reference level

† This is an alternative specific constant reflecting the respondents' preference for the fixed alternative of the fingerprick-test over Device B. Participants were informed that a fingerprick-test should be done four times a day, requires high effort to check, does not result in skin irritation or redness, will show your glucose levels, doesn't have an alarm and costs €25 per month.

Note: Due to non-linearity of the attributes, all were effects-coded, enabling the direct comparison of the estimates. The sum of the effect coded attributes is zero, and therefore the coefficient of the reference category can be easily calculated and the relative importance of the reference categories of the attributes can be compared with one another, and so that the alternative specific constants have independent interpretation signifying the average utility for that alternative.

**Table 3. Swing weighting preference weights, calculated through both point allocation and the rank order centroid (ROC) method.**

| Attribute | WCM | All respondents (n = 459) | | | Saw DCE first (n = 233) | | | Saw SW first (n = 226) | | |
|---|---|---|---|---|---|---|---|---|---|---|
| | | Mean | SD | SE | Mean | SD | SE | Mean | SD | SE |
| Cost | PA | 0.17 | 0.13 | 0.01 | 0.18 | 0.14 | 0.01 | 0.16 | 0.13 | 0.01 |
| | ROC | 0.17 | 0.13 | 0.01 | 0.18 | 0.13 | 0.01 | 0.16 | 0.13 | 0.01 |
| Precision | PA | 0.16 | 0.12 | 0.01 | 0.17 | 0.12 | 0.01 | 0.16 | 0.12 | 0.01 |
| | ROC | 0.16 | 0.11 | 0.01 | 0.17 | 0.11 | 0.01 | 0.16 | 0.11 | 0.01 |
| Pricks | PA | 0.15 | 0.11 | 0.01 | 0.15 | 0.11 | 0.01 | 0.15 | 0.12 | 0.01 |
| | ROC | 0.18 | 0.11 | 0.01 | 0.19 | 0.11 | 0.01 | 0.17 | 0.11 | 0.01 |
| Information | PA | 0.14 | 0.12 | 0.01 | 0.15 | 0.12 | 0.01 | 0.14 | 0.11 | 0.01 |
| | ROC | 0.13 | 0.11 | 0.01 | 0.15 | 0.12 | 0.01 | 0.14 | 0.11 | 0.01 |
| Effort | PA | 0.13 | 0.09 | >0.00 | 0.12 | 0.07 | >0.00 | 0.14 | 0.11 | 0.01 |
| | ROC | 0.13 | 0.10 | >0.00 | 0.11 | 0.09 | 0.01 | 0.14 | 0.11 | 0.01 |
| Skin Irritation | PA | 0.12 | 0.09 | >0.00 | 0.12 | 0.09 | 0.01 | 0.12 | 0.09 | 0.01 |
| | ROC | 0.12 | 0.10 | >0.00 | 0.11 | 0.10 | 0.01 | 0.12 | 0.10 | 0.01 |
| Alarms | PA | 0.12 | 0.11 | 0.01 | 0.11 | 0.09 | 0.01 | 0.13 | 0.12 | 0.01 |
| | ROC | 0.11 | 0.11 | 0.01 | 0.11 | 0.10 | 0.01 | 0.12 | 0.11 | 0.01 |

SW = swing-weighting; DCE = discrete choice experiment; WCM = weight calculation method; SD = standard deviation; SE = standard error of mean; PA = point allocation; ROC = rank order centroid.

**Comparison of weight distribution between the DCE and SW.** For the DCE, the proportion of attribute importance is very different for all the attributes (Fig 3). Contrastingly, all attributes in the SW, received between 12–17% of the designated importance. The DCE had a 14.9-fold difference between proportional importance of the most and least important attribute, while the SW had a 1.4-fold difference.

The two attributes with the highest importance were cost and precision, respectively, for both the DCE and SW using point allocation, but the relative weight of costs was much higher in the DCE. For the DCE, the following order of attributes based on their relative importance weight was: skin irritation, fingerpricks, effort, alarms, and glucose information, respectively. For the SW using point allocation, these were fingerpricks, glucose information, effort, skin irritation, and alarms, respectively. The relative weights of all these attributes differed significantly between the two methods.

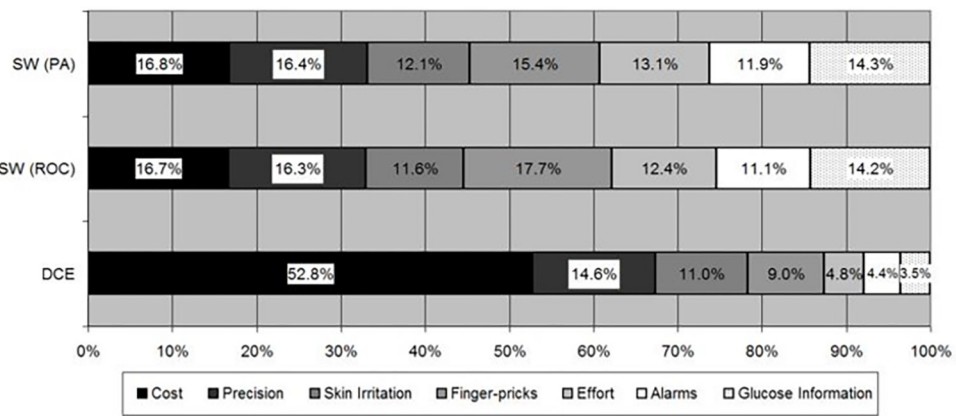

**Fig 3. Proportion of attribute importance relative to sum of all attributes' importance (DCE and SW calculated through both ROC and PA.** DCE = discrete choice experiment; SW = swing weighting; PA = point allocation; ROC = rank order centroid method.

**Sensitivity analyses.** The sensitivity analysis using the ROC instead of the point allocation method to calculate SW, revealed slight differences in the importance of attributes (Table 3 and Fig 3). Fingerprick frequency was the most important attribute (mean = 0.18; s.d. = 0.11), followed by cost (mean = 0.17; s.d. = 0.13), with alarm being the least important attribute (mean = 0.11; s.d. = 0.11). There was a 1.5 fold difference between the proportional importance of the lowest and highest attribute, compared to the 14.9 -fold difference in the DCE. The ROC method for determining weights still achieved very little difference in the relative weights given to the seven attributes, (Fig 3).

The (generalised) ordered logit model (see Appendix V in S1 File) also indicated that there were significant differences in the respondent-level rankings of the attributes between the DCE and SW. Fingerprick frequency was more likely to be ranked as the highest attribute in the DCE rather than in the SW and alarms and precision were more likely to be ranked among the bottom-ranked attributes in DCE rather than in the SW. Other attributes showed significant differences in rank order between the two methods as well.

## Discussion

Both the DCE and SW point allocation identified that cost was the most important attribute for diabetes patients when selecting a glucose-monitoring device. Preference outcomes in both methods were unaffected by the order in which they were completed. However, the weights derived from the SW were almost evenly distributed regardless of a calculation through point allocation or ROC method. The SW point allocation had a 1.4-fold difference between the most and least important attribute, while the DCE had a 14.9-fold difference. The DCE was better received by participants, and obtained more detailed insights for all attribute-levels, making it the preferred method over the SW in this case study.

To the best of our knowledge, this is the first study directly comparing the outcomes of a DCE and a SW task in which the relative weights were able to be compared. Previous research compared the two methods, but were unable to directly compare the outcomes [17].

The small difference between the mean attribute weights in the SW warrants further discussion. As the point allocation part of the SW task was a direct rating, participants essentially created the weights themselves thus negating the need for researchers to convert rankings to surrogate weights. Incorporating point allocation into SW is often praised for being a simple way to elicit the relative valuation of the attributes by allowing respondents to directly report this valuation for each attribute ("providing information on relative importance, whilst remaining relatively uncomplicated" [33]). However, there remains some uncertainty about how the point allocation should be administered. The direct ranking task we used did not force participants to trade-off when allocating points to the different attributes. In this way, there was no cost to valuing one attribute over another like there is in a DCE. Other SW techniques ask participants to designate a proportion of 100 points to each attribute, meaning all attribute weights must add up to 100 [15, 16, 34]. While this results in a trade-off between the attribute valuation this type of task has been found to be less reliable. For our case study, the added complexity of trading off points between seven attributes was deemed to be an unnecessary numerical burden if participants had to monitor the total sum score while awarding points. Additional complexity may result in random responses, or responses becoming unresponsive to small differences in points. Therefore, in this study, participants could award any points out of 100, and their weights were calculated out of their total sum. One issue that has previously been raised with this type of rating method is the poor discriminatory power resulting from insufficient variability in the point allocation [35]. This tends to muddle the differences in the valuation and was evidenced in our study where over 55% of participants allocated the same number of points to at least two attributes.

The sensitivity analysis of the ROC method was important to identify whether the even distribution of weights was only a product of the point allocation methodology [36]. The same phenomenon occurred with the ROC verifying that it is likely a characteristic of SW analysis itself. Crucially, there were small differences found for the most important attributes. The ROC is often criticised for the extreme weights it places on higher-ranked attributes with minimal difference between the weights of lower-ranked attributes [34]. The findings of this study support the conclusion that point allocation as a robust weight calculation method than ROC and should be used in future SW studies.

Neither weight-calculation technique of the SW gave as specific an insight as the DCE, which forces choices between pairwise comparisons. DCEs 'decompose' treatment or medical product alternatives into specific attributes describing the element that are most influential to patient decisions. This makes it possible to estimate preferences for more levels per attribute than only the best and worst level which are used in SW [35]. An additional benefit of DCEs are the ability to assess preference heterogeneity using mixed logit models. The value of this method was found in our study results which demonstrated strong preference heterogeneity for most device attributes. Finally, the outcomes of a DCE can be used for more than just relative weights of attributes thus the outcomes of one study can be applied to a wider range of applications [37].

The DCE was better received by participants than the SW, regardless of the order completed, or the level of health literacy or health numeracy reported by the patient. Accuracy of preference measurements is highly dependent on patient understanding. Whether respondents started with DCE or SW did not significantly affect preference outcomes in either method. This suggests that the combination of two methods did not create overwhelming cognitive burden or study fatigue, or that there was not a significant ordering effect regarding the way experimental materials were presented.

Previous literature comparing DCE and SW comprehensibility has been lacking; however, it has been theorised that direct pairwise comparisons in a DCE are easier for patients than a direct numerical assessment of relative value present in SW despite the increased cognitive burden attached to assessing multiple attributes concurrently [14, 38]. Additionally, evidence suggests that rating methods are not observed to be easy cognitive tasks due to the involvement of a predetermined numerical scale, and the complexity of applying it against multiple attributes [39]. Essentially, it is easier to say which of two attributes is more important, rather than trying to quantify how much more important it is [14]. From the researcher's perspective, the DCE may appear more complex compared to the SW in terms of design and analysis, but respondents view the DCE as the simpler method to understand and complete. A SW is a viable alternative in cases when the number of attributes cannot be feasibly integrated into a DCE [7], or when a sample size is too small for a DCE, such as in the case of rare diseases [8, 37].

## Limitations

The length of the survey and number of screening questions could have contributed to the drop-out rate within the panel data, or created a higher cognitive burden. Due to confidentially agreements, reminder e-mails could not be sent and a non-response analysis could not be conducted. It took on average 19.2 minutes to complete the survey, which was faster than expected, and could be a limitation of the study if participants did not spend sufficient time reading the instructions. Participants who completed the survey faster than 70% of the mean response time were excluded (n = 41) due to their speed decreasing the chance of them having read all elements of the survey. A sensitivity analysis revealed their exclusion bettered the model fit and decreased left-right bias.

About 93.3% reported a "high" or "intermediate" level of education (defined in Appendix IV in S1 File) meaning there was an underrepresentation of participants with a low level of education. Approximately 18.3% of all respondents already used a CGM or FGM, and 54.4% used fingerpricking, while 27.2% used neither. Individual uptake probabilities for the most-preferred device compared to fingerpricking varied between CGM/FGM-users and finger-pricking-users (13.7% versus 33.0%, respectively).

The listed order of the attributes remained the same for each choice task of the DCE, with precision listed first and cost last, which means that participants could have ignored attributes in the middle when scanning the choice tasks. However, lexicographic behaviour (i.e. always opting for the best level of one attribute) was very low in the dataset, with sensitivity analysis revealing little difference if these (n = 19) participants were removed from the sample. The SW always had its attributes randomised during the ranking exercise.

The feedback for understanding and completing the SW exercise did not distinguish between ranking the attributes and point allocation, so therefore we cannot know which part of the SW the participants found the most difficult. This could have helped understanding whether the point allocation was a valuable addition to the exercise.

The ordered logit was conducted with the ranking information from the SW exercise only. This analysis could not be performed using data from the point allocation, due to 55% of participants allocating an equal number of points to at least two attributes in the point allocation.

## Implications for future research

Future research should examine DCE and SW in more head-to-head studies with different populations, different medical products treatments, and different decision contexts in order to examine if the same weight distribution occurs. Variations of the SW point allocation should be examined, reducing the number of attributes, forbidding attributes being allocated the same number of points, or forcing all attributes to add to 100. More studies comparing the point allocation system to the ROC method would also help conclude whether point allocation adds meaningful quantitative insight into preferences, or merely adds cognitive burden to respondents.

## Conclusions

This study compared a DCE with SW by eliciting preference for glucose-monitoring devices in a population of 459 diabetes patients. Both methods identified that cost was the most important attribute when selecting a device, followed by the precision of the device. However, the weights derived from the SW, regardless of a calculation through point allocation or ROC method, were almost evenly distributed between the attributes. The DCE was better received by participants, and generated larger weight differences between each attribute level, making it the more informative method in our case study. This method comparison provides further evidence of the degree of method suitability and trustworthiness of these methods for measuring preferences for decision-making. Further research should compare these methods in different disease areas and decision-contexts.

## Supporting information

**S1 File.**
(DOCX)

## Acknowledgments

The authors would like to thank the PREFER consortium, Erasmus Choice Modelling Centre (ECMC), the employees of SurveyEngine, including Alex Parkman, Ludwig von Butler, and Mandy van Dijk. Thank you as well to Anna Śliwińska (Polskie Stowarzyszenie Diabetyków), Rimke C. Vos (Department of Public Health and Primary Care, LUMC–Campus Den Haag).

## Author Contributions

**Conceptualization:** Chiara Whichello, Ian Smith, Jorien Veldwijk, G. Ardine de Wit, Maureen P. M. H. Rutten- van Molken, Esther W. de Bekker-Grob.

**Data curation:** Chiara Whichello, Ian Smith.

**Formal analysis:** Chiara Whichello.

**Funding acquisition:** Chiara Whichello.

**Investigation:** Chiara Whichello, Ian Smith.

**Methodology:** Chiara Whichello.

**Project administration:** Chiara Whichello, Ian Smith.

**Resources:** Chiara Whichello.

**Supervision:** Jorien Veldwijk, G. Ardine de Wit, Maureen P. M. H. Rutten- van Molken, Esther W. de Bekker-Grob.

**Validation:** Chiara Whichello.

**Visualization:** Chiara Whichello.

**Writing – original draft:** Chiara Whichello, Ian Smith.

**Writing – review & editing:** Chiara Whichello, Ian Smith, Jorien Veldwijk, G. Ardine de Wit, Maureen P. M. H. Rutten- van Molken, Esther W. de Bekker-Grob.

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
