## [Decision Letter · Decision Letter 0]

19 Jul 2022

PONE-D-21-37787Discrete choice experiment versus swing-weighting: A head-to-head comparisonPLOS ONE

Dear Dr. Whichello,

Thank you for submitting your manuscript to PLOS ONE. After careful consideration, we feel that it has merit but does not fully meet PLOS ONE’s publication criteria as it currently stands. Therefore, we invite you to submit a revised version of the manuscript that addresses the points raised during the review process.

We look forward to receiving your revised manuscript.

Kind regards,

Richard Huan XU

Academic Editor

PLOS ONE

Journal Requirements:

2. Please include a complete ethics statement in the Methods section, including the name of the IRB, the approval number, and a statement on whether the study was approved or whether approval was waived. We note that the current statement in the Appendices appears to both state that the study was approved and that approval was waived - please specify which one of these is correct. Please also clarify how participants provided consent."

Reviewers' comments:

Reviewer's Responses to Questions

**Comments to the Author**

1. Is the manuscript technically sound, and do the data support the conclusions?

Reviewer #1: Yes

2. Has the statistical analysis been performed appropriately and rigorously? 

Reviewer #1: Yes

3. Have the authors made all data underlying the findings in their manuscript fully available?

Reviewer #1: No

4. Is the manuscript presented in an intelligible fashion and written in standard English?

Reviewer #1: Yes

5. Review Comments to the Author

Reviewer #1: Thanks the authors for this interesting paper. This study compares discrete choice experience and swing weighting in quantifying relative importance of attributes of glucose monitoring device. The study design and implementation are robust, while there are a few places need further clarification or additional details.

1. I would suggest that the title should specify this DCE/SW comparison was made for glucose monitoring device among patients with diabetes.

2. Could the authors show the initial 12 attributes derived from the interviews in the supplementary file and the primary reasons why some of them were excluded? It would help the audience understand how the seven attributes were selected.

3. It seems the order of the attributes shown in the DCE choice sets was not randomized for different respondents, which may lead to position bias (the cost and precision, the last and the first attribute in the choice tasks, happened to be the two most important attributes identified from DCE). Please clarify whether the randomization was done, and why if not randomized, and how it may affect the outcome (in comparison with SW which the attributes were randomized).

4. What is the response rate of the survey? Was the randomization to three blocks and the order of DCE and SW stratified by their age, type of diabetes, and/or current glucose monitor usage, etc.? Could the authors also report whether there was difference in key characteristics of the patients across DCE blocks and the orders of DCE/SW?

5. For DCE part, how many percentages of respondents in the analytical dataset (n=459) consistently chose the alternatives on the left-hand or right-hand side in all 12 choice tasks? This may also indicate that the respondents did not actually focus on the choice tasks.

6. Is there any difference in relative importance of the attributes (both methods as shown in Figure 3) for patients with different glucose-monitoring device usage and health literacy/numeracy? This could show whether the DCE and SW results are sensitive to whether having relevant knowledge and experience, which can help other researchers to interpret their results of these methods.

7. Could the authors provide more details in the sensitivity analysis on respondent-level rankings of the attributes using the ordered logit models, e.g. how the respondent-level ranking of attributes was defined in DCE, the model specification, etc.? In addition, the sensitivity analysis finding showed that fingerpick frequency were more likely to rank the highest in DCE than SW, while Figure 3 shows that proportion of importance of fingerpicks is 9.0% in DCE, which is lower than 15.4% and 17.7% in SW, could the authors explain this inconsistency?

8. The relative importance identified by ROC and point allocation seems to be very similar in Table 3 and Figure 3. Could the authors clarify how they drew the conclusion that “The findings of this study support the conclusion that point allocation is a more robust weight calculation method than ROC (line 372-373)”, apart from the evidence from previous studies?

6. PLOS authors have the option to publish the peer review history of their article (what does this mean?). If published, this will include your full peer review and any attached files.

Reviewer #1: No

---

## [Author Response · Author response to Decision Letter 0]

6 Feb 2023

Dear PlosOne Editors, 

We would like to re-submit our PONE-D-21-37787 paper entitled ‘Discrete choice experiment versus swing-weighting: A head-to-head comparison’ to PlosOne, because we believe this article could be of great interest for you and your readers. We sincerely appreciate the comments from your reviewer, and we would like to integrate and address them. 

1) Comment: I would suggest that the title should specify this DCE/SW comparison was made for glucose monitoring device among patients with diabetes.

i. Response: Thank you for the suggestion, we have re-titled to “Discrete choice experiment versus swing-weighting: A head-to-head comparison of diabetic patient preferences for glucose-monitoring devices”

2) Comment: Could the authors show the initial 12 attributes derived from the interviews in the supplementary file and the primary reasons why some of them were excluded? It would help the audience understand how the seven attributes were selected.

i. Response: In step 3, the list of 12 attributes were rated according to relevance, completeness, non-redundancy, operationality, and preferential independency by the research team. This process resulted in five attributes being were removed (based on failing these criteria) or being combined with other, similar attributes. This resulted in a final list of seven attributes which were used for the DCE and SW (Table 1).

3) Comment: It seems the order of the attributes shown in the DCE choice sets was not randomized for different respondents, which may lead to position bias (the cost and precision, the last and the first attribute in the choice tasks, happened to be the two most important attributes identified from DCE). Please clarify whether the randomization was done, and why if not randomized, and how it may affect the outcome (in comparison with SW which the attributes were randomized).

i. Response: This is correct that attribute randomization was not implemented for the DCE – and this is addressed and the impact discussed in paragraph 3 of the limitations section. As discussed, lexicographic behaviour was very low in the dataset, therefore the impact was minimal. 

4) Comment: What is the response rate of the survey? Was the randomization to three blocks and the order of DCE and SW stratified by their age, type of diabetes, and/or current glucose monitor usage, etc.? Could the authors also report whether there was difference in key characteristics of the patients across DCE blocks and the orders of DCE/SW?

i. Response: Of 5,620 invited participants, 500 completed the survey, indicating a response rate of 8.9%, which we have added to the supplemental file. The randomization to the elicitation methods (50:50) or the three DCE blocks (33:33:33) or elicitation method were randomly allocated across the entire sample, and there were no meaningful sociodemographic or clinical characteristics between the assignments. 

5) Comment: For DCE part, how many percentages of respondents in the analytical dataset (n=459) consistently chose the alternatives on the left-hand or right-hand side in all 12 choice tasks? This may also indicate that the respondents did not actually focus on the choice tasks.

i. As reported in the section “Discrete choice experimental results”, there was a slight left-right bias, and this is also reported in the last row of Table 2, with the coefficient mean 0.359 (p<0.01). However, this is very common in DCE surveys in languages that read from left to right, and in line with other DCE studies. 

6) Comment: Is there any difference in relative importance of the attributes (both methods as shown in Figure 3) for patients with different glucose-monitoring device usage and health literacy/numeracy? This could show whether the DCE and SW results are sensitive to whether having relevant knowledge and experience, which can help other researchers to interpret their results of these methods.

i. Response: It is possible that preferences can vary between subgroups, such as high or low health numeracy/literacy. However, we feel that exploring this would be outside the scope of this paper and not quite aligned with the research objectives – we want to prioritise comparing the DCE and SW using a single, identical sample since this paper was methodological in focus and comparing the relative-attribute importance of various sociodemographic or clinical subgroups would shift the focus.

7) Comment: Could the authors provide more details in the sensitivity analysis on respondent-level rankings of the attributes using the ordered logit models, e.g. how the respondent-level ranking of attributes was defined in DCE, the model specification, etc.? In addition, the sensitivity analysis finding showed that fingerpick frequency were more likely to rank the highest in DCE than SW, while Figure 3 shows that proportion of importance of fingerpicks is 9.0% in DCE, which is lower than 15.4% and 17.7% in SW, could the authors explain this inconsistency?

i. Response: Thank you for this comments, and we have provided more detail about the ordered logic model under the heading “sensitivity analysis”. Figure 3 shows the relative attribute importance of a single attribute as a proportion of all attribute importance – essentially the proportion of decisions (i.e. choice task decisions) that were driving preferences and the attribute with the largest influence on patients’ choices of treatments. The differences between DCE and SW are discussed in detail in the section “comparison of weight distribution between the DCE and SW”. The sensitivity analysis in Appendix VI does not necessarily show that finger-prick frequency would be “more likely” to rank overall higher in the DCE than the SW: merely that, for example, this attribute had a 47% chance of being ranked #1 in a DCE, compared to a 19% chance in the SW. The sensitivity analysis was strictly comparing individual rankings (1 to 7) against other individual rankings (1 to 7). 

8) Comment: The relative importance identified by ROC and point allocation seems to be very similar in Table 3 and Figure 3. Could the authors clarify how they drew the conclusion that “The findings of this study support the conclusion that point allocation is a more robust weight calculation method than ROC (line 372-373)”, apart from the evidence from previous studies?

i. Response: Thank you for this helpful comment. You are correct that this is an overstatement and we have adjusted it to be more in line with our previous statements: that there are indeed slight differences between point allocation and the ROC method, but point allocation was as robust as ROC for this study. 

Thank you for your attention, and looking forward to hearing from you, 

Chiara Lauren Whichello, PhD

---

## [Decision Letter · Decision Letter 1]

1 Mar 2023

PONE-D-21-37787R1Discrete choice experiment versus swing-weighting: A head-to-head comparison of diabetic patient preferences for glucose-monitoring devicesPLOS ONE

Dear Dr. Whichello,

Thank you for submitting your manuscript to PLOS ONE. After careful consideration, we feel that it has merit but does not fully meet PLOS ONE’s publication criteria as it currently stands. Therefore, we invite you to submit a revised version of the manuscript that addresses the points raised during the review process.

We look forward to receiving your revised manuscript.

Kind regards,

Richard Huan XU

Academic Editor

PLOS ONE

Journal Requirements:

Reviewers' comments:

Reviewer's Responses to Questions

**Comments to the Author**

1. If the authors have adequately addressed your comments raised in a previous round of review and you feel that this manuscript is now acceptable for publication, you may indicate that here to bypass the “Comments to the Author” section, enter your conflict of interest statement in the “Confidential to Editor” section, and submit your "Accept" recommendation.

Reviewer #1: (No Response)

2. Is the manuscript technically sound, and do the data support the conclusions?

Reviewer #1: Yes

3. Has the statistical analysis been performed appropriately and rigorously? 

Reviewer #1: Yes

4. Have the authors made all data underlying the findings in their manuscript fully available?

Reviewer #1: No

5. Is the manuscript presented in an intelligible fashion and written in standard English?

Reviewer #1: Yes

6. Review Comments to the Author

Reviewer #1: Thanks the authors to provide revisions to the manuscript which addresses most of the comments.

There is one more comment from me. For sensitivity analysis for attribute rankings between DCE and SW, while the authors have added a few details to its method, it is still unclear how respondent-level ranking of attributes was derived from DCE. Could the authors tell more specifically how to use DCE marginal utility (which I assumed refers to model estimates in Table 2) to find out the attribute rankings of each individual respondent?

7. PLOS authors have the option to publish the peer review history of their article (what does this mean?). If published, this will include your full peer review and any attached files.

Reviewer #1: No

---

## [Author Response · Author response to Decision Letter 1]

17 Mar 2023

Dear PlosOne Editors, 

We would like to re-submit our PONE-D-21-37787 paper entitled ‘Discrete choice experiment versus swing-weighting: A head-to-head comparison of diabetic patient preferences for glucose-monitoring devices’ to PlosOne, because we believe this article could be of great interest for you and your readers. We sincerely appreciate the minor revision requested from your reviewer, and we would like to integrate and address it in this resubmission. 

1) Reviewer #1: Thanks the authors to provide revisions to the manuscript which addresses most of the comments. There is one more comment from me. For sensitivity analysis for attribute rankings between DCE and SW, while the authors have added a few details to its method, it is still unclear how respondent-level ranking of attributes was derived from DCE. Could the authors tell more specifically how to use DCE marginal utility (which I assumed refers to model estimates in Table 2) to find out the attribute rankings of each individual respondent?

i. Response: Thank you for this comments, and we have provided more detail the heading “sensitivity analyses” for the methodology of (generalized) ordered logit model. We hope that this explanation provides greater clarification: 

“The respondent-level ranking of the attributes in the DCE and the SW were identified by determining how each individual participant ranked the attributes, either by participant rankings for the SW, or a ranked marginal utility for the DCE method. These individual ranking were then compared using a (generalised) ordered logit model. For the SW, these were individual-specific ranking outputs derived from Step 1 of the SW method, ranking the attributes from 1 to 7. For the DCE, the individual-specific rankings of each attribute were determined by first examining the patient uptake rates for the most preferred device (high precision, zero fingerpricks, low effort, low skin irritability, 25 euro, plain information, no alarm) replicating the same device hypothetically created during the SW exercise (i.e. the swing from ‘worst’ to ‘best’ attributes). As detailed above, the systematic utility is defined as an additive function consisting of marginal utilities. Therefore, the coefficients for each corresponding attribute-level for each individual were then ranked from lowest to highest and given a corresponding value (1 to 7). Each participant’s DCE rankings were then compared against their SW rankings through the ordered logit model, and determining the probability that an attribute had of being ranked 1-7 in either the DCE or the SW.”

We sincerely hope that this submission will be accepted upon this resubmission. Thank you for your attention, and looking forward to hearing from you, 

Chiara Lauren Whichello, PhD

---

## [Editor Report · Decision Letter 2]

21 Mar 2023

Discrete choice experiment versus swing-weighting: A head-to-head comparison of diabetic patient preferences for glucose-monitoring devices

PONE-D-21-37787R2

Dear Dr. Whichello,

We’re pleased to inform you that your manuscript has been judged scientifically suitable for publication and will be formally accepted for publication once it meets all outstanding technical requirements.

Kind regards,

Richard Huan XU

Academic Editor

PLOS ONE
---

## [Editor Report · Acceptance letter]

10 Apr 2023

PONE-D-21-37787R2 

Discrete choice experiment versus swing-weighting: A head-to-head comparison of diabetic patient preferences for glucose-monitoring devices 

Dear Dr. Whichello:

I'm pleased to inform you that your manuscript has been deemed suitable for publication in PLOS ONE. Congratulations! Your manuscript is now with our production department. 

Kind regards, 

on behalf of

Dr. Richard Huan XU 

Academic Editor

PLOS ONE